# Characterization of QTLs for Seedling Resistance to Tan Spot and Septoria Nodorum Blotch in the PBW343/Kenya Nyangumi Wheat Recombinant Inbred Lines Population

**DOI:** 10.3390/ijms20215432

**Published:** 2019-10-31

**Authors:** Pawan Kumar Singh, Sukhwinder Singh, Zhiying Deng, Xinyao He, Zakaria Kehel, Ravi Prakash Singh

**Affiliations:** 1International Maize and Wheat Improvement Center (CIMMYT), Apdo. Postal 6-641, México 06600, D.F., Mexico; Sukh.Singh@cgiar.org (S.S.); deng868@163.com (Z.D.); x.he@cgiar.org (X.H.); Z.Kehel@cgiar.org (Z.K.); R.Singh@cgiar.org (R.P.S.); 2State Key Laboratory of Crop Biology, Cooperation Innovation Center of Efficient Production with High Annual Yield of Wheat and Corn, Shandong Agricultural University, Taian 271018, China

**Keywords:** Genetics, *Triticum aestivum*, host-pathogen interaction, *Pyrenophora tritici-repentis*, *Parastagonospora nodorum*

## Abstract

Tan spot (TS) and Septoria nodorum blotch (SNB) induced by *Pyrenophora tritici-repentis* and *Parastagonospora nodorum*, respectively, cause significant yield losses and adversely affect grain quality. The objectives of this study were to decipher the genetics and map the resistance to TS and SNB in the PBW343/Kenya Nyangumi (KN) population comprising 204 F_6_ recombinant inbred lines (RILs). Disease screening was performed at the seedling stage under greenhouse conditions. TS was induced by *P. tritici-repentis* isolate MexPtr1 while SNB by *P. nodorum* isolate MexSN1. Segregation pattern of the RILs indicated that resistance to TS and SNB in this population was quantitative. Diversity Array Technology (DArTs) and simple sequence repeats (SSRs) markers were used to identify the quantitative trait loci (QTL) for the diseases using inclusive composite interval mapping (ICIM). Seven significant additive QTLs for TS resistance explaining 2.98 to 23.32% of the phenotypic variation were identified on chromosomes 1A, 1B, 5B, 7B and 7D. For SNB, five QTLs were found on chromosomes 1A, 5A, and 5B, explaining 5.24 to 20.87% of the phenotypic variation. The TS QTL on 1B chromosome coincided with the pleiotropic adult plant resistance (APR) gene *Lr46/Yr29/Pm39*. This is the first report of the APR gene *Lr46/Yr29/Pm39* contributing to TS resistance.

## 1. Introduction

Wheat (*Triticum aestivum* L.) is a cereal grain, originating from the Near East but now being grown all over the globe, occupying the largest land area compared to other commercial food crops, which, however, makes it face multiple biotic and abiotic stresses. The fungal leaf spotting diseases complex comprising tan spot (TS, caused by *Pyrenophora tritici-repentis* (Died.) Drechs.), Septoria nodorum blotch (SNB, *Parastagonospora nodorum* (Berk.) Quaedvlieg, Verkley & Crous), Septoria tritici blotch (STB, *Zymoseptoria tritici* (Desm.) Quaedvlieg & Crous) and spot blotch (SB, *Cochliobolus sativus* (Ito & Kuribayashi) Drechs. Ex Dastur) cause major damage to wheat production globally. Among these diseases, TS and SNB in recent decades have observed significant increases globally [1]. On average, TS and SNB cause generally 5 to 10% yield losses; however, under conditions favourable for disease development, yield losses of up to 50% have been reported [2,3]. Murray and Brennan [4] reported that in Australia the three most devastating pathogens were *P. tritici-repentis*, *Puccinia striiformis* and *P. nodorum*, with an average annual losses caused by the diseases induced by these pathogen amounting to $212 million, $127 million and $108 million, respectively. Serious yield losses due to TS and SNB occur due to a reduction in the photosynthetic area of leaves resulting in reduced grain fill, lower test weight, kernel shriveling, and lower number of kernel per head [3,5]. TS and SNB also render red or dark smudges and black points to the infected kernels, significantly downgrading wheat quality [5,6].

Isolates of the fungus *P. tritici-repentis* (Died.) Drechs. [anamorph *Drechslera tritici-repentis* (Died.) Shoem.] differ in virulence and have been race classified. Susceptibility of wheat to *P. tritici-repentis* is shown via the development of tan necrosis and/or extensive chlorosis symptoms depending on the race of the pathogen and the susceptibility in the host [5]. Presently, eight races of *P. tritici-repentis* have been identified worldwide based on their virulence pattern on a set of wheat differential cultivars involving Glenlea, 6B-365, 6B-662 and Salamouni/Erik [7] and on the presence of genes coding for specific host-selective toxins (HSTs) [8]. Race structure in the fungus *P. nodorum* (Berk.) Quaedvlieg, Verkley & Crous (anamorph *Stagonospora nodorum* (Berk.) E. Castell. & Germano) is not universally accepted; significant variation in aggressiveness among isolates is observed. *P. nodorum* produces multiple HSTs that contribute to variation in aggressiveness and for infection to occur, the host must produce the target receptor protein recognized by the toxin [9].

Genetic analysis of wheat-*P. tritici-repentis* interactions have revealed that host resistance ranges from qualitative [10,11,12,13,14] to quantitative control [15,16,17,18,19,20]. Some studies utilizing quantitative trait loci (QTL) analysis reported resistance to TS as polygenic, but observed that previously reported major genes frequently underline these QTLs [18,20,21]. Isolates of the fungus *P. tritici-repentis* are known to produce at least three HSTs; Ptr ToxA [22], Ptr ToxB [23] and Ptr ToxC [24] that interact with products of specific host sensitivity genes located on chromosome arm 5BL [10,11], 2BS [25] and 1AS [24], respectively, to cause disease. However, Friesen, et al. [26] observed that insensitivity to HST Ptr ToxA accounted for only a part of the resistance to the fungus and pointed out that Ptr ToxA was a virulence factor instead of a pathogenicity factor. Similar findings were observed by other studies [16,27], indicating that there may be other mechanisms than HSTs involved in disease development including non-toxin factors, host non-specific toxins, and unknown host-specific toxins [7].

For SNB, although several reports have identified a single dominant gene conferring resistance [28,29], there are studies that testified the genetics of adult plant resistance is quantitative [20,30,31,32,33,34]. So far, there have been 15 HSTs found, of which, eight (SnToxA, SnTox1, SnTox2, SnTox3, SnTox4, SnTox5, SnTox6 and SnTox7) were characterized from *P. nodorum*, which interacted with host genes such as *Tsn1*, *Snn1*, *Snn2*, *Snn3*, *Snn4*, *Snn5*, *Snn6* and *Snn7* [35,36,37]. In order to understand the inheritance of SNB resistance, genetic mapping and QTL analysis have been researched [9,20,32,38,39]. In the last decade, there were some reports on QTLs for seedling resistance and adult plant resistance [39,40]. The identified QTLs of seedling resistance were located on 1B, 2B, 2D, 3A, 4A, 4B, 5A, 5B, 5D, 6A, and 7A chromosomes [35,41,42]. Among them, some genomic regions contributed to adult plant resistance. For example, the QTL on 5BL at the *Tsn1* locus for seedling resistance was in the same marker interval as the QTL for flag leaf resistance [35], similar to those on 2D, 4B and 5A chromosomes [35,40]. Some QTLs for flag leaf resistance have been found to be associated with the host genes, such as on 1B with *Snn1*, on 2D with *Snn2*, and on 5B with *Tsn1*.

One of the major objectives of the CIMMYT’s wheat breeding program is to develop superior germplasm with durable resistance to multiple diseases including TS and SNB. Identification, characterization and transfer of new resistance genes from different resistant sources are always underway to enhance the level and diversity of resistance. These diverse genetic resources potentially possessing multiple resistance genes are then incorporated into breeding programs to develop durable multiple disease resistant germplasm. The mega variety PBW343 is a high yielding, widely adapted and very popular cultivar in India, which is resistant to TS and SNB but susceptible to stem rust Ug99. Kenya Nyangumi (KN) shows adult plant resistance to Ug99 but is susceptible to TS and SNB. Primarily, the population PBW343/KN was developed to decipher the adult plant resistance to Ug99; however, since this population segregates for TS and SNB as well, this study was initiated to determine and map the resistance to TS and SNB.

## 2. Results

### 2.1. Disease Phenotyping

The checks for the experiments gave consistent reaction for inoculation with *P. tritici-repentis* race 1, with Glenlea being susceptible-necrotic, 6B-365 being susceptible-chlorotic, and Erik and 6B-662 being resistant. The statistics showed a significant (*p* < 0.01) difference between resistant (Erik and 6B-662) and susceptible (Glenlea and 6B-365) checks and no significant difference between the two experiments (*p* = 0.63) for TS (Table 1). Almost 70% of the TS variability within the experiment was achieved by the RILs. The broad-sense heritability was observed to be 68.72% (Table 2). The parents of RILs showed contrasting reaction to TS, with the resistant parent PBW343 showing a BLUP (best linear unbiased predictions) value of 1.90 while the susceptible parent KN showed a BLUP value of 3.71 (Figure 1). The RILs showed normal distribution with some transgression observed, indicative of the quantitative nature of disease resistance. The BLUP mean of all the 204 RILs was 2.78 with a range of 1.55 to 4.50.

For SNB, there was also a significant (*p* < 0.01) difference between resistant (Erik and 6B-662) and susceptible (Glenlea and 6B-365) checks and no significant difference between the two experiments (*p* = 0.64) (Table 1). The broad-sense heritability was 63.62%. Moreover, the distribution of SNB resistance in this population was similar to that of TS resistance (Figure 1). The mean of this population was 2.52 with a range of 1.41 to 4.13. There was also transgressive distribution for SNB in this population.

The results showed significant variation in TS and SNB resistance among the parental genotypes and the RILs of the PBW343/KN population used in the present study, suggesting its suitability for conducting QTL analysis.

### 2.2. Molecular Analysis

Of the 1383 markers used for genotyping the RIL population, 475 were polymorphic, including 442 DArT (Diversity arrays technology) makers, 31 SSR (Simple sequence repeat) markers and 2 STS (Sequence-Tagged Sites) markers. Thirty-eight linkage groups were generated and assigned to wheat chromosomes using the available maps with DArT, SSR and STS markers. The linkage map covered 2208 centimorgan (cM) with an average density of one marker per 4.65 cM, which was then used to map QTLs for TS and SNB.

### 2.3. QTL Analysis

ICIM analysis revealed seven significant additive QTLs for TS resistance, explaining 2.98 to 23.32% of the phenotypic variation. These QTLs were distributed on 1A, 1B, 5B, 7B, and 7D chromosomes (Table 3 and Figure 2). The three major QTLs on 1A, 5B, and 7B with their respective percent variation explained (PVE) of 19.00%, 23.32%, and 11.75% (Table 3) were designated as *QTs.cim-1A.2*, *QTs.cim-5B.2*, and *QTs.cim-7B*, respectively. All the favorable alleles were contributed by PBW343, except for *QTs.cim-7B* and *QTs.cim-1B*. Two epistatic QTLs were detected, i.e., *QTs.cim-5B.2/QTs.cim-1B.2* and *QTs.cim-6A/QTs.cim-1A.2*, but the PVEs were very small. These results indicated that the additive effects were more important than the epistatic effects for TS resistance in this population. The QTL on 7D was a new QTL for TS. In addition, the QTL *QTs.cim-1B* contributed by the susceptible parent KN was located between *CSLV46-1B* and *wPt-1070*, a region harboring the pleiotropic gene *Lr46/Yr29/Pm39* for wheat rusts and powdery mildew.

Five QTLs for SNB were found on chromosomes 1A, 5A and 5B, explaining 5.24% to 20.87% of phenotypic variation (Table 4 and Figure 2). Of these, three with PVE greater than 10% were designated as *QSnb.cim-1A.2*, *QSnb.cim-5A*, and *QSnb.cim-5B.2*, with the resistance alleles all from PBW343. Of the two minor QTLs, *QSnb.cim-1A.1* was contributed by KN and *QSnb.cim-5B.1* by PBW343. It should be noted that *QSnb.cim-5B.2* was located on the same marker interval as *QTs.cim-5B.2*, very close to *XFCP393* for *Tsn1*. Despite controlling different diseases, these two QTLs were placed on the same marker interval.

### 2.4. Interaction among Important QTLs

The RILs with important QTLs and their combinations were identified according to the genotypes of flanking markers. The mean disease severities of RILs with single or combination of QTLs were found to be reduced in the presence of favorable alleles at the corresponding loci (Table 5 and Table 6). For instance, the 84 RILs with PBW343 allele showed 0.79 lower TS severity compared to the 58 with ‘KN’ allele at *QTs.cim-5B.2*, assessed by the nearest marker *XFCP393* (Table 5). The same situation applied to *QSnb.cim-5B.2* with an SNB severity reduction of 1.00 (Table 6). In addition, significant difference was also found between lines with and without *QTs.cim-1A.2*. It was very interesting that lines with any of the three above-mentioned QTLs always showed significantly reduced disease when compared to those without the corresponding QTLs, regardless the presence of other QTLs. Most of the QTL combinations were positively validated. TS resistance QTLs showed highly significant interactions in reducing the TS severity when in combination with *QTs.cim-5B.2* on 5BL (Table 5), as did SNB resistance QTLs. The most significant reduction (1.63) of TS disease severity was observed in RILs positive for the combination of *QTs.cim-5B.2* and *QTs.cim-1A.2*, indicating the additive effects of these two QTLs in reducing disease severity. Similarly, severity was reduced from 3.18 to 1.58 with *QTs.cim-5B.2*, *QTs.cim-7B* and *QTs.cim-1A.2*. While for SNB, the combination of three QTLs showed the maximum significant reduction from 3.11 to 1.70, followed by the combination of *QSnb.cim-5B.2* and *QSnb.cim-5A*. The QTLs from chromosome 5BL reduced the TS and SNB by 0.79 and 1, respectively.

## 3. Discussion

Since its inception, CIMMYT’s primary goal has been to develop superior wheat germplasm incorporating broad-spectrum disease resistance, homogeneity in agronomic traits that impart high yields, superior quality, adaptability and stability. One of the strategies of developing disease resistant germplasm involves breeding for adult plant resistance (APR), which is associated with non-hypersensitive reaction, race-nonspecific resistance, quantitatively inherited and possibly effective against multiple diseases. The focus on APR was based on the fact that stacking multiple genes of small effects leads to high levels of resistance, which is highly durable due to its quantitative resistance nature [43]. Previous reports have indicated that the race-nonspecific resistance genes *Lr34* and *Lr46* confer broad-spectrum resistance to biotrophic diseases like rusts and powdery mildew and hemi-biotrophic disease like spot blotch [44].

TS and SNB are two major components of the destructive leaf-spotting diseases complex of wheat. Resistant varieties are regarded as the most efficient and effective strategies of disease management, but the information on resistance genes or QTLs is a prerequisite for development of broad genetic based durable resistant cultivars. Additionally, to hasten the process and progress of marker-assisted selection (MAS) it is paramount that new resistance genes are precisely tagged. Many studies on QTL mapping for TS and SNB have been reported [9,19,35,45,46,47], but most of them were aimed at single diseases. The RIL population used in this study was not only characterized for TS and SNB, but also for stem rust [48]. It is interesting to note that some important QTLs found on 5BL were associated with all three diseases, which implied multi-disease resistance QTLs.

In this research, we identified seven TS QTLs, contributed mostly by PBW343, of which *QTs.cim-5B.2* on chromosome 5BL showed the biggest phenotypic effects. This QTL was reported in previous research and corresponds to *Tsn1* [14,20,46,49]. Cheong et al. [15] and Singh et al. [18] also found QTLs corresponding or close to the location of *Tsn1* (*Tsr1*) with PVE of 39% and 18%, respectively. Molecular markers closely linked to this locus have been identified and are available for MAS [50]. Another major QTL was detected on 1AS, where *QTsc.ndsu-1A* was initially reported by Faris et al. [27] and validated by Effertz et al. [21,24] to be associated with chlorosis in both adult plants and seedlings. Sun et al. [51] identified *QTs.ksu-1AS* in this region, whose position coincided with *Tsc1*, and its effect was attributed to the Tsc1-Ptr ToxC interaction. Faris et al. [46] identified molecular markers linked to *Tsc1*, of which several STS makers could be useful in MAS. QTLs on 1BS, 7BS and 7DS chromosomes with lower PVE were detected in the current study, some of which might be the same as reported previously. On 1BS, Faris and Friesen [17] identified a race-nonspecific QTL, explaining from 13 to 29% of the phenotypic variation. On 7BS, Faris, et al. [52] identified a race-nonspecific QTL with PVE ranging from 5 to 8%. On 7DS, Faris and Friesen [17] detected a minor QTL, but it was specific for race 3, suggesting that the 7DS QTL found in our study might be a new one.

There were also many reports on QTLs for SNB resistance in both seedling and adult plant. Most of them were located on 1BS, 2AS, 2BS, 2D, 3BS, 4BL, 5AL, 5BL, 5DL, 6AL and 7BL chromosomes [35], of which, the 5BL QTL was the most frequently reported, for which four out of six reports were on seedling resistance. Similarly, two QTLs on 5BL were identified in the present study, with one being a major and the other a minor QTL. The major QTL was close to the *Tsn1* locus according to their linked markers. Additionally, a major QTL was found on 5AS, which has not been reported so far and thus may be a new QTL for SNB resistance. Two more possible new QTLs were detected on the 1A chromosome, one of which showed major effects.

In this population, we detected QTLs on 5BL for both TS and SNB resistance, and Singh et al. [48] also found a QTL, *QSr.cim-5B* for stem rust resistance, in a similar region. These results indicated that the 5BL QTL region could be very important for multi-disease resistance, and further studies are needed to see if it confers resistance to other diseases. PCR-based markers linked to this locus are available for MAS [48]. Another possible multi-disease resistant locus in this population was *QTs.cim-1B* for TS resistance, which was closely linked to the resistance gene *Lr46/Yr29/Pm39* for rusts and powdery mildew, suggesting that *Lr46/Yr29/Pm39* might also be able to contribute to TS resistance, but their mechanism of interaction needs to be further studied. Despite this, the functional marker for this locus is available for MAS [53].

The findings of this study suggest that the genomic regions contributing to TS and SNB resistance may also confer resistance to other diseases, such as rusts and powdery mildew resistance [47,54], which perhaps indicates there are some race/pathogen-nonspecific resistance genes. Therefore, utilization of those broad-spectrum resistance genes/QTLs is recommended for breeding superior multi-disease resistant wheat cultivars, whereas the disease-specific QTLs identified in the current study are useful for breeding programs aimed at TS and SNB resistance.

## 4. Materials and Methods

### 4.1. Plant Material

A segregating population of 204 F_6_ RILs was developed at CIMMYT, Mexico, from a cross between cultivars PBW343 and KN. The male parent KN, pedigree TEZANOS-PINTOS-PRECOZ//SELKIRK-ENANO*6/LERMA-ROJO-64/3/AFRICA-MAYO48/4/KENYA-SWARA/KENYA-4500, is from Kenya and the pedigree of the female parent PBW343 is ND/VG9144//KAL/BB/3/YACO/4/VEE#5. The RIL population was used for genetic analysis and mapping of resistance to TS and SNB.

### 4.2. Experimental Design

The three experiments for TS and SNB reaction were conducted under controlled conditions in a greenhouse, and each experiment comprised two replications arranged in a completely randomized design. Each replication contained a complete set of RILs planted in trays. The experimental unit comprised four plants per RIL, and parents and checks (Erik, Glenlea, 6B-365, and 6B-662) were included in each experiment to verify the inoculation process and the race of the inoculum used.

### 4.3. Disease Screening

Spore inoculum of *P. tritici-repentis* was produced using the method of Lamari and Bernier [55] with minor modifications. Briefly, dried mycelial plugs of isolate MexPtr1 (race 1) of 0.5-cm-diameter were placed on 10-cm Petri plates with V8-potato dextrose agar (V8-PDA) (150 mL V8-juice, 10 g PDA, 10 g agar, 3 g CaCO_3_, and 850 mL distilled water). The plates were incubated in the dark at room temperature (20 to 22 °C) for six days. Subsequently, the plates were flooded with sterile distilled water, and the mycelium was flattened with the bottom of a sterile test tube. Excess water was drained from the dishes, which were then incubated under continuous light at 22–24 °C for one day followed by one day in the dark in an incubator at 16 °C to induce conidiophore and conidia production, respectively. For spore collection, about 25 mL sterile distilled water was added to the plates and a camel-hair brush was used to dislodge conidia. Spore concentration was measured with a haemocytometer and adjusted to 4 × 10^4^ conidia/mL before inoculation.

A spore suspension of *P. nodorum* isolate MexSN-4 was used as inoculum for SNB experiments using a method modified from Ma and Hughes [28]. The isolate was grown in 10-cm-diameter Petri dishes with V8-agar medium (150 mL V8-juice, 15 g agar, 1.5 g CaCO_3_, and 850 mL distilled water) at room temperature and under continuous illumination. Fresh cultures of the *P. nodorum* isolate were made by transferring stock cultures onto the surface of fresh V8-agar medium. Seven days later, pink pycnidia of 0.5–1.0 mm diameter appeared on the surface of V8-agar media, and subsequently a spore suspension was prepared by firstly flooding the plates with sterile distilled water and then gently brushing the medium surface with a camel-hair brush so that the spores were released. The obtained spore suspension was filtered through two layers of cheesecloth to eliminate mycelium. The spore concentration was finally adjusted to 1 × 10^7^ spores/mL prior to inoculation.

The seedlings were grown and maintained in the greenhouse at a temperature of 22/18 °C (day/night) with a photoperiod of 16 h. The seedlings were watered and fertilized as needed. Inoculation took place at the two-leaf stage, when the conidial suspension was sprayed onto seedlings until runoff, with a hand sprayer. After inoculation, the seedlings were kept in a mist chamber for 24 h under continuous misting and then moved back onto benches in the same greenhouse. Disease evaluation was performed at eight days after inoculation, based on a 1 to 5 scale [54,55].

### 4.4. Molecular Marker Analysis

DNA extraction followed the procedure described by Singh and Bowden [56]. A Nano-Drop ND1000 spectrophotometer (Thermo Fisher Scientific Inc., Waltham, MA, USA) was used for DNA quantification. For DArT genotyping, 500–1000 ng of high quality DNA in 50–100 ng/µL was shipped to Triticarte Pty. Ltd., Canberra, Australia (www.triticarte.com.au) for genome profiling [57,58]. Allele calling was made as present (1) or absent (0), and the overall call rate was approximately 95% and Q values for marker quality were above 80% for most markers. The markers were designated with ‘w’ as the prefix if the clone was from wheat; ‘t’ if it was from triticale and ‘r’ if it was from rye libraries.

### 4.5. Linkage Mapping and QTL Analysis

ICIMapping software was used in the current study for constructing linkage maps with the default parameter settings [59], and a total of 1383 markers including 1350 DArT markers, 31 SSR markers and 2 STS markers were used for linkage mapping in the MAP functionality of the software. QTL mapping was performed with the same software using the ICIM algorithm in the BIP functionality [59]. An LOD score of 2.5 was chosen as a threshold for declaring significant QTL.

### 4.6. Statistical Analysis

Analysis of variance for lesion type disease score in the experiments was performed using Genstat 15th edition (Hempstead, UK). The best linear unbiased predictors (BLUPs) are appropriate for estimation of test line effects as this method has a minimum mean squared error property [60]. A linear mixed model was fitted to the TS and SNB variables to estimate the BLUPs for each of the RILs and parents. Checks were assumed to be fixed effects. The broad-sense heritability was defined as *H*^2^ = $\sigma_{g}^{2}$/$\sigma_{p}^{2}$, where $\sigma_{g}^{2}$ is the genotypic variance and $\sigma_{p}^{2}$ is the phenotypic variance. 

## Figures and Tables

**Figure 1 ijms-20-05432-f001:**
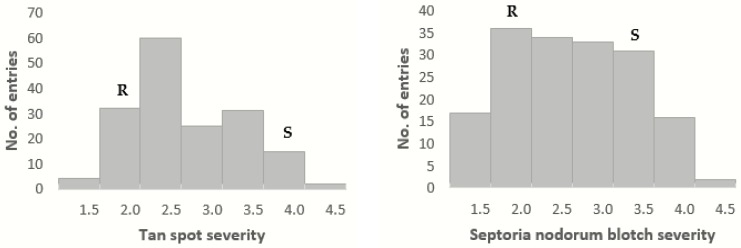
Histogram of best linear unbiased predictions (BLUPs) of 204 recombinant inbred lines (RILs) derived from the cross PBW343 (R-Parent)/KN (S-Parent) after inoculation with *P. tritici-repentis* isolate MexPtr1 and *P. nodorum* isolate MexSN4.

**Figure 2 ijms-20-05432-f002:**
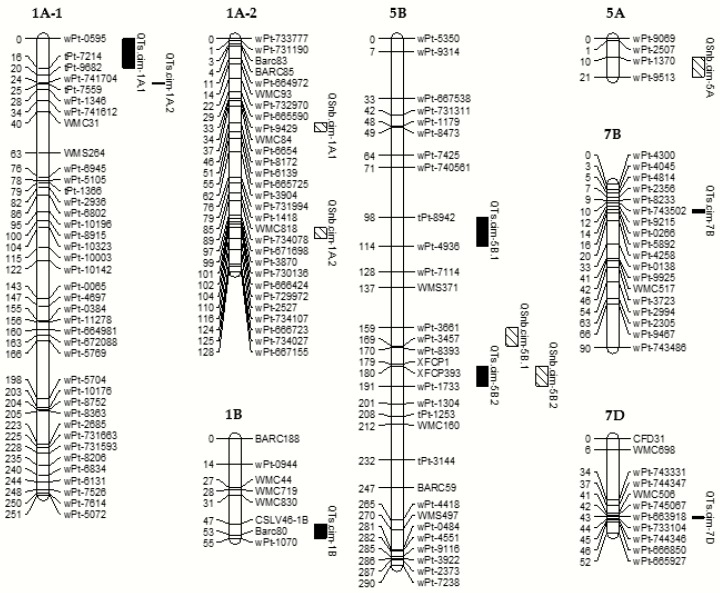
Linkage groups with significant QTLs for seedling resistance to tan spot and Septoria nodorum blotch in the PBW343/KN population. Refer to Table 3 and Table 4 for the naming of the linkage groups.

**Table 1 ijms-20-05432-t001:** Analysis of variation for fixed effects in TS and SNB experiments of the PBW343/Kenya Nyangumi population.

Experiment	Source	Degree of Freedom	*F* Statistic	*p* Values
TS	Check	3	107.14	<0.001
	Exp	1	0.53	0.634
SNB	Check	4	73.59	<0.001
	Exp	2	0.52	0.642

**Table 2 ijms-20-05432-t002:** Variance components of TS and SNB in 204 RILs of the PBW343/Kenya Nyangumi population.

Experiment	Source	Variance Component	% Total Variance	Standard Error
TS	RILs	0.390	68.720	0.042
	Exp.Rep	0.001	0.123	0.001
	RIL.Exp.Rep	0.056	9.857	0.030
	Residuals	0.121	21.299	0.029
	Total	0.568	100	
SNB	RILs	0.4737	63.618	0.0514
	Exp.Rep	0.0035	0.47	0.0039
	RIL.Exp.Rep	0.1344	18.05	0.0335
	Residuals	0.133	17.862	0.0314
	Total	0.7446	100	

**Table 3 ijms-20-05432-t003:** QTLs for TS resistance in the population of PBW343 (−)/Kenya Nyangumi (+).

QTL	Chr.	Pos. *^a^*	Flanking Markers		LOD	PVE (%) *^b^*	Add *^c^*
			Left	Right			
QTs.cim-5B.1	5B	112	wPt-8942	wPt-4936	4.33	2.98	−0.11
QTs.cim-5B.2	5B	180	XFCP393	wPt-1733	17.73	23.32	−0.213
QTs.cim-7B	7B	14	wPt-0266	wPt-5892	35.27	11.75	0.23
QTs.cim-1B	1B	52.8	CSLV46-1B	wPt-1070	4.83	3.00	0.11
QTs.cim-7D	7D	43	wPt-663918	wPt-733104	8.86	5.78	−0.15
QTs.cim-1A.1	1A	0	wPt-0595	tPt-7214	6.02	3.69	−0.12
QTs.cim-1A.2	1A	25	wPt-741704	tPt-7559	23.18	19.00	−0.29

*^a^* position in cM; *^b^* percent variation explained; *^c^* additive effect.

**Table 4 ijms-20-05432-t004:** QTLs for SNB resistance in the population of PBW343 (−)/Kenya Nyangumi (+).

QTL	Chr.	Pos. *^a^*	Flanking Markers		LOD	PVE (%) *^b^*	Add *^c^*
			Left	Right			
QSnb.cim-5B.1	5B	168	wPt-3661	wPt-3457	4.88	8.00	−0.20
QSnb.cim-5B.2	5B	180	XFCP393	wPt-1733	11.38	19.51	−0.32
QSnb.cim-1A.1	1A	47	wPt-8172	wPt-6139	5.70	5.24	0.17
QSnb.cim-1A.2	1A	104	wPt-729972	wPt-2527	12.05	11.45	−0.26
QSnb.cim-5A	5A	10	wPt-1370	wPt-9513	11.24	10.69	−0.25

*^a^* position in cM; *^b^* percent variation explained; *^c^* additive effect.

**Table 5 ijms-20-05432-t005:** Comparison of RILs having different tan spot QTLs.

QTL	Absence	Presence	Difference	*p* Value	R^2^
QTs.cim-5B.2	2.94	2.15	0.79	<0.0001	0.31
QTs.cim-7B	2.75	2.49	0.26	0.0127	0.04
QTs.cim-1B	2.83	2.49	0.34	0.0045	0.06
QTs.cim-1A.2	2.85	2.23	0.62	<0.0001	0.20
QTs.cim-5B.2 + QTs.cim-7B	2.80	2.23	0.57	<0.0001	0.33
QTs.cim-7B + QTs.cim-1B	3.14	2.44	0.70	0.0002	0.15
QTs.cim-5B.2 + QTs.cim-1B	2.80	2.26	0.55	<0.0001	0.36
QTs.cim-1B + QTs.cim-1A.2	2.81	2.57	0.25	<0.0001	0.30
QTs.cim-7B + QTs.cim-1A.2	2.70	2.27	0.43	<0.0001	0.22
QTs.cim-5B.2 + QTs.cim-1A.2	3.26	1.63	1.63	<0.0001	0.59
QTs.cim-5B.2 + QTs.cim-7B + QTs.cim-1B	2.85	2.48	0.36	<0.0001	0.41
QTs.cim-5B.2 + QTs.cim-7B + QTs.cim-1A.2	3.18	1.58	1.60	<0.0001	0.59
QTs.cim-7B + QTs.cim-1B + QTs.cim-1A.2	2.95	2.71	0.24	<0.0001	0.37
QTs.cim-5B.2 + QTs.cim-7B + QTs.cim-1B + QTs.cim-1A.2	3.07	2.67	0.40	<0.0001	0.66

**Table 6 ijms-20-05432-t006:** Comparison of RILs having different *Septoria nodorum* blotch QTLs.

QTL	Absence	Presence	Difference	*p* Value	R^2^
QSnb.cim-5B.2	3.00	2.01	1.00	<0.0001	0.43
QSnb.cim-1A.2	2.53	2.60	0.07	0.5562	0.00
QSnb.cim-5A	2.62	2.40	0.22	0.0630	0.02
QSnb.cim-5B.2 + QSnb.cim-1A.2	2.98	1.99	0.99	<0.0001	0.40
QSnb.cim-5B.2 + QSnb.cim-5A	3.15	1.86	1.28	<0.0001	0.44
QSnb.cim-1A.2 + QSnb.cim-5A	2.58	2.25	0.33	0.1222	0.04
QSnb.cim-5B.2 + QSnb.cim-1A.2 + QSnb.cim-5A	3.11	1.70	1.41	<0.0001	0.49

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
