# Peer review of "Characterization of QTLs for Seedling Resistance to Tan Spot and Septoria Nodorum Blotch in the PBW343/Kenya Nyangumi Wheat Recombinant Inbred Lines Population"

_ijms, 2019, doi:10.3390/ijms20215432_

Round 1

Reviewer 1 Report

This manuscript used diversity array technology (DArTs), simple sequence repeat sequence (SSRs) markers and inclusive composite interval mapping (ICIM) to analyze the quantitative trait loci (QTL) for the diseases in the PBW343/Kenya Nyangumi (KN) population comprising 204 F6 recombinant inbred lines(RILs). Some significant QTLs have been identified to explain phenotypic variation.This study has certain research significance in the application of wheat disease-resistant molecular breeding, but the manuscript still has many shortcomings. Specific Suggestions are as follows:

The reference cited in the introduction is relatively early, which does not reflect the latest research progress. In addition, the manuscript does not cite a reference of the past five years, that is not convincing. The main research content of this manuscript is QTL analysis, while the description of this method in 4.5 only cites one article to illustrate, which is obviously not enough and needs to provide more details. Line 98: please explain the abbreviation of the first time, and carefully check the same problem in many parts of the article. Line 116: table 1 is not clear. Line 261: document format is not uniform. Figure 1: the figure is not clear enough, the data is obscured, and the bar chart lacks the title of horizontal and vertical coordinates.

Author Response

The reference cited in the introduction is relatively early, which does not reflect the latest research progress. In addition, the manuscript does not cite a reference of the past five years that is not convincing.

Response: Suggestion accepted and now more recent publications on the two diseases have been cited in the revised manuscript.

The main research content of this manuscript is QTL analysis, while the description of this method in 4.5 only cites one article to illustrate, which is obviously not enough and needs to provide more details.

Response: More details on functionality of the software used and parameter setting is added in the revised manuscript.

Line 98: please explain the abbreviation of the first time, and carefully check the same problem in many parts of the article.

Response: Suggestion taken and addressed.

Line 116: table 1 is not clear.

Response: The title has been modified to express clearly the aim.

Line 261: document format is not uniform.

Response: This has been re-formatted.

Figure 1: the figure is not clear enough, the data is obscured, and the bar chart lacks the title of horizontal and vertical coordinates.

Response: The figure has been re-drawn.

Reviewer 2 Report

The paper is devoted to a very important problem, namely the identification of QTLs for seedling resistance to tan spot and septoria nodorum blotch a – two major components of the destructive leaf-spotting diseases complex of wheat.

The plant material used was PBW343/Kenya Nyangumi (KN) population comprising 204 F6 recombinant inbred lines. Using statistical methods such as analysis of variance and inclusive composite interval mapping the authors identified seven significant additive QTLs for TS resistance explaining 2.98 to 23.32% of the phenotypic variation and five QTLs for SNB explaining 5.24 to 20.87% of the phenotypic variation.

The performed analysis allowed the authors to conclude that there may exist some race/pathogen-nonspecific resistance genes as the genomic regions contributing to TS and SNB resistance may also confer other diseases resistance. Such wide range resistance QTLs may be utilized for breeding multi-disease resistant cultivars.

I suggest to accept the paper in current form.

Author Response

Response: No comments from Reviewer that needs to be addressed.

Round 2

Reviewer 1 Report

The author has revised the manuscript as requested.